# Cytochrome P450 Surface Domains Prevent the β-Carotene Monohydroxylase CYP97H1 of *Euglena gracilis* from Acting as a Dihydroxylase

**DOI:** 10.3390/biom13020366

**Published:** 2023-02-15

**Authors:** Thomas Lautier, Derek J. Smith, Lay Kien Yang, Xixian Chen, Congqiang Zhang, Gilles Truan, Nic D Lindley

**Affiliations:** 1Singapore Institute of Food and Biotechnology Innovation (SIFBI), Agency for Science, Technology and Research (A*STAR), Singapore 138669, Singapore; 2Toulouse Biotechnolgy Institute, Université de Toulouse, CNRS, INRAE, INSA, 31077 Toulouse, France; 3CNRS@CREATE, 1 Create Way, #08-01 Create Tower, Singapore 138602, Singapore

**Keywords:** cytochrome P450, carotenoids, β-cryptoxanthin, protein engineering, regioselectivity, asymmetric catalysis

## Abstract

Molecular biodiversity results from branched metabolic pathways driven by enzymatic regioselectivities. An additional complexity occurs in metabolites with an internal structural symmetry, offering identical extremities to the enzymes. For example, in the terpene family, β-carotene presents two identical terminal closed-ring structures. Theses cycles can be hydroxylated by cytochrome P450s from the CYP97 family. Two sequential hydroxylations lead first to the formation of monohydroxylated β-cryptoxanthin and subsequently to that of dihydroxylated zeaxanthin. Among the CYP97 dihydroxylases, CYP97H1 from *Euglena gracilis* has been described as the only monohydroxylase. This study aims to determine which enzymatic domains are involved in this regioselectivity, conferring unique monohydroxylase activity on a substrate offering two identical sites for hydroxylation. We explored the effect of truncations, substitutions and domain swapping with other CYP97 members and found that CYP97H1 harbours a unique N-terminal globular domain. This CYP97H1 N-terminal domain harbours a hydrophobic patch at the entrance of the substrate channel, which is involved in the monohydroxylase activity of CYP97H1. This domain, at the surface of the enzyme, highlights the role of distal and non-catalytic domains in regulating enzyme specificity.

## 1. Introduction

Carotenoids (>1100 members) are natural pigments widely distributed in plants, animals, algae and microbes [1]. Across living organisms, carotenoids are involved in different functions, such as light harvesting in the photosystem, oxidative stress tolerances, intra-species communication or ecological functions [2]. In animals, carotenoids play important roles, such as in the light-absorbing prosthetic group of rhodopsin in the retina [3]. However, animals do not produce de novo carotenoids and obtain them through their diets. This need for external sourcing increases interest in understanding plant carotenoid biosynthesis in order to unlock biotechnological approaches. Carotenoids belong to the vast terpene family, harbouring conjugated double bounds [4]. The molecular biodiversity among carotenoids requires regioselective enzymes to drive the metabolic flux into specific metabolic branches. The carotenoid family is built by the iterative addition of isoprene-based blocks. Reaching an aliphatic chain of 40 carbons, phytoene is the substrate for a succession of internal desaturations and/or cyclisations of the extremities. These two types of reactions are competitive, leading to a branched metabolic pathway. This long aliphatic chain can also be oxygenated by dioxygenases, with cleavage possibilities, leading to carotenoids with asymmetric carbon chain lengths. Due to the axial symmetry of the carotene chain, the intermediates offer equivalent moieties to the enzymes, which are then able to act several times on the same initial metabolite backbone. Deciphering the enzymatic mechanism involved in the functionalisation of specific intermediates is needed to understand the roots of this molecular diversity. In this study, we used the degree of hydroxylation of β-carotene as an example. A succession of desaturations of phytoene produces lycopene. One of the two extremities of lycopene can be cyclised into a ꞵ-cycle, leading to γ-carotene, i.e., monocyclised β-carotene (Figure 1). The remaining linear extremity of γ-carotene can then be cyclised by the same lycopene cyclase, leading to double β-cyclised lycopene, i.e., β-carotene. Different lycopene cyclases exist, and the remaining linear extremity can be cyclised by an ε cyclase, leading to α-carotene, which harbours a β-cycle and an ε-cycle. The α- and β-carotene cycles can be the substrates to various hydroxylases and ketolases from non-heme hydroxylases or cytochrome P450 families, particularly from the cytochrome P450 97 family (CYP97) [5]. Considered one of the oldest plant-specific CYP clans, the CYP97 family is one of the five cytochrome P450 families present across several kingdoms, found in green algae, land plants and several protists; the CYP97 family is involved in xanthophyll’s synthesis, i.e., oxygenated carotenes [6,7].

Non-heme hydroxylases, such as CrtZ or BCH1, are described as dihydroxylases on β-carotene. They produce zeaxanthin, composed of two identical hydroxylated β-cycles, with no accumulation of β-cryptoxanthin, the monohydroxylated intermediate [8]. The enzymatic oxidation occurs firstly on one of the two cycles, at position 3 (or 3′). As the molecule extremities are symmetric, the monohydroxylated β-carotene can flip and enter a second time into the hydroxylase, offering its opposite cycle for a second hydroxylation, producing zeaxanthin. However, whether the hydroxylase acts as a dimer, with each monomer assuming one hydroxylation, remains unknown [8]. In the non-heme hydroxylase BCH1 from *Arabidopsis thaliana*, the truncation of the first 129 amino acids at the N-terminal favours the accumulation of monohydroxylated β-carotene. It has been suggested that this domain triggers homodimerisation, but biochemical demonstration has not yet been established [8].

In addition to the non-heme hydroxylase active on β-carotene, cytochrome P450 carotene hydroxylases are mainly involved in the sequential double hydroxylation of α-carotene, composed of an ε-cycle and a β-cycle. The cytochrome P450 (CYP) carotene hydroxylases are divided into two classes, active either on the carotene β-cycle or on the carotene ε-cycle. In the lutein pathway, in which the scaffold is α-carotene, two types of P450 are involved and act in concert: in *Daucus carota*, CYP97A3 hydroxylates the α-carotene on the β-cycle, leading to zeinoxanthin, and then CYP97C1 hydroxylates the ε-cycle of zeinoxanthin to produce lutein [9]. The length of the substrate channel in CYP97A3 from *Arabidopsis thaliana* suggests that the entire C40 α-carotene is embedded inside one enzyme [10]. With a substrate fully embedded inside the cytochrome P450, enzymatic dimers for simultaneous hydroxylations on each extremity of the carotene is therefore not a likely hypothesis. Regarding the cytochrome P450 activity on the β-carotene, several cytochromes from the CYP97 family, such as CYP97A3 from *A. thaliana* or CYP97A4 from *Oriza sativa,* have been noted to act as β-carotene dihydroxylases [11,12]. An exception among the cytochrome P450 97 family is CYP97H1 from the photosynthetic protist *Euglena gracilis*, which acts as a β-carotene monohydroxylase, producing β-cryptoxanthin [13]. What are the CYP97H1 domains involved in this monohydroxylase activity that prevent the classic second hydroxylation of β-cryptoxanthin?

This study aimed to establish if a specific domain of CYP97H1 was involved in the monohydroxylase activity of this enzyme. To that end, we conducted protein engineering on the specific domains present in CYP97H1 compared to the rest of the CYP97 members. The effects of deletion, substitution or domain swapping of these specific domains were evaluated by expressing the mutants in a β-carotene producing *E. coli* strain and by quantifying the production of mono- and dihydroxylated carotene.

## 2. Materials and Methods

### 2.1. Strain and Plasmid Construction

Strains for the upper part of the carotene pathway were based on our designed astaxanthin strain [14]. The mevalonate pathway genes were spread in four plasmids: p15A-*spec-hmgS-atoB-hmgR* (L2-8) (*atoB, hmgS* and truncated *hmgR*) and into p15A-*cam-mevK-pmk-pmd-idi* (L2-5) (*mevk, pmk, pmd* and *idi*). The lycopene pathway genes were cloned into p15A-*kan-crtEBI-isp* (*crtEBI* and *ispA*). The last module controls cyclisation (crtY) and hydroxylation (various hydroxylases with their redox partner) (Table 1). The protein sequences of the truncated version and the chimera are detailed in Appendix A. Under T7 variants, gene expression was induced by isopropyl β-d-1-thiogalactopyranoside (IPTG). These platform strains have led to production of various carotenoids and derived products with yields above 500 mg/L, or even over 25 g/L for some terpenoids [14,15].

### 2.2. P450 Structure Modelling and Substrate Docking

Alignment of the CYP97 sequences from UniProt was performed through ClustalW [16,17]. 3D models of CYP97H1 from *Euglena gracilis* were generated using Swiss-MODEL [18], YASARA [19] and AlphaFold2 [20] through CoLabFold [21], which ended up being the model used in this article. AlphaFold was initially used to calculate 5 models of the full-length sequence, but the first 184 and the last 20 residues were modelled with high disorder, due to a lack of sequence data in this region. Given this result, a truncated CYP97H1 sequence starting from G185 and ending at P716 was remodelled using the pdb70 template option, with 48 recycles and amber relaxation. The best model was superposed against the structure of *A. thaliana* CYP97A3 (PDB 6J95) (RMSD 0.85Å across 341 residues) and heme coordinates were then copied straight from the structure to give the final model for the study.

Idealised 3D coordinates of β-carotene, β-cryptoxanthin, rubixanthin and zeaxanthin were obtained from the NCBI PubChem website (https://pubchem.ncbi.nlm.nih.gov, (accessed on 9 June 2022) in SDF format. Both protein and substrate coordinates were processed using AutoDockTools [22] to add partial charges and polar hydrogens. Docking was performed using AutoDock VINA [23,24] using a docking grid of x = 28 Å, y = 40 Å and z = 50 Å centred around the binding pocket, with a search exhaustiveness of 10. All results were viewed in PyMOL [25]. 

### 2.3. Culture of the E. coli Strains

The strains were cultivated in 2 mL of 2XPY (20 g/L peptone, 10 g/L yeast extract and 10 g/L NaCl) supplemented with 10 g/L glycerol, 75 mM 4-(2-hydroxyethyl)-1-piperazineethanesulfonic acid (HEPES) and Tween 80 0.5%, as previously described [14]. Initial growth was achieved at 37 °C/250 rpm until OD_600 nm_ reached ~0.6–0.8, induced by 0.05 mM of IPTG. 0.5 mM of delta aminolevulinic acid, the heme precursor, was added to the inducer, and the production phase was conducted at 16, 20, 25, 28 or 37 °C for 48 h. To maintain the four plasmids, the culture was supplemented with antibiotics (34 μg/mL chloramphenicol, 50 μg/mL kanamycin, 50 μg/mL spectinomycin and 100 μg/mL ampicillin). Chemicals were purchased from Sigma-Aldrich.

### 2.4. Extraction of Carotenoids

Briefly, 20–40 µL microbial cultures (depending on the expected content of cellular carotenoids) were collected and centrifuged. After a water wash, the pellets were resuspended in 20 µL of water and carotenoids were extracted by the addition of 180 µL of acetone and homogenisation for 50 min at 50 °C. After centrifugation at 14,000× *g* for 10 min, the supernatant was directly injected into HPLC.

### 2.5. Quantification of Carotenoids

Carotenoids were separated using an Agilent 1290 Infinity II UHPLC System and detected with a Diode Array Detector (DAD). The analytical method was adapted from a previous protocol [14]. Briefly, 5 μL of extracted carotenoids in acetone were injected into the column Agilent ZORBAX RRHD Eclipse Plus C18 2.1 × 50 mm, 1.8 μm using a flow rate of 1 mL/min. The eluent gradient started with 60% methanol:water 4:1 and 40% acetonitrile for 1 min, followed by increasing acetonitrile from 40% to 80% in the following 3 min. This condition continued for 11 min. The entire analysis finished at 15 min. 

Lycopene, β-carotene, β-cryptoxanthin, rubixanthin and zeaxanthin were purchased from CaroteNature, Switzerland. As the carotenoid pathway is a highly branched pathway, with several close oxygenated intermediates or isomers, lycopene, β-carotene, β-cryptoxanthin, rubixanthin and zeaxanthin HPLC peaks have been identified using coelution standard retention time coupled to carotenoid-specific visible spectra (using the ratio between the three peaks showing the maximum absorbance, specific to each carotene). As zeaxanthin, β-cryptoxanthin and rubixanthin share molecular similarities, mass spectrometry fragmentation profiles were also used to confirm each peak detected by absorbance. Lycopene and zeaxanthin stock solution concentrations were quantified by dissolving a known mass in hexane. Due to standard quantity limitation, β-carotene and β-cryptoxanthin standards were dissolved in hexane; these stock solution concentrations were calculated using the absorption coefficients A^1%^_450 nm_ 2500 and 2356, respectively (Sigma Chemical and [26]). HPLC standard curves were established for the five standards using peak areas at 450 nm of serial dilution. The peak areas of each extracted compound were used to calculate the carotenoid concentrations. 

LC-MS/MS analysis was performed using an Agilent UPLC1290 coupled with an Agilent 6540 UHD accurate-mass quadrupole time-of-flight (Q-TOF) mass spectrometer. Separation of extracts was carried out on an Agilent ZORBAX RRHD Eclipse Plus C18 (2.1 × 50 mm, 1.8 μm) at a flow rate of 0.5 mL/min. Both mobile phases A (water) and B (methanol) consisted of 0.1% formic acid. Initially, the run started at 90% mobile phase B for 2 min, then linearly increased to 100% B in 1.5 min, and stayed at 100% B for 8 min. The total run time was 10.5 min, followed by a post-run of 1 min.

The typical QTOF operating parameters were as follows: positive ionisation mode; sheath gas nitrogen flow, 12 L/min at 295 °C; drying gas nitrogen flow, 8 L/min at 275 °C; nebuliser pressure, 30 psi; nozzle voltage, 1.5 kV; capillary voltage, 4 kV. Lock masses in positive ion mode: purine ion at *m/z* 121.0509 and HP-921 ion at *m/z* 922.0098. 

The identification of carotenoid compounds was performed by comparing the retention time (RT), mass-to-charge ratio (*m*/*z*) value and characteristic fragments with standards (summarised in Table 2). During electrospray ionisation (ESI) in positive ion mode, the molecular ion was observed as [M]^+•^. All carotenoids also showed a neutral loss of 92 Da, which is consistent with the literature [27].

## 3. Results

### 3.1. In Silico Identification of the Structural Specificities of CYP97H1 Compared to CYP97 Members

The protein engineering study started with aligning members of the CYP97 family to identify specific CYP97H1 characteristics in the primary sequence and model comparison (Figure 2). In the sequence alignment, heme and oxygen binding sites were identified as highly conserved. Loops and helices involved in the redox partner interaction were also well-defined (Figure 2A). Compared to other members of the CYP97 family, for which sequences are available in the Uniprot database, CYP97H1 has a unique internal deletion of six residues located between the residues G423 and T424. This deletion is located in the F′-G′ loop, known to play a drawbridge role in substrate specificity, which can be analysed by using chimera targeting on this portion [28].

Moving forward on the alignment analysis (Figure 2), the N-terminal of CYP97H1 appears to harbour a specific region. From residue M1 to L116, the sequence is predicted to be a peptide signal for plastid targeting, with two putative transmembrane helices. Other members of the CYP97 family also present a plastid-targeting N-terminal sequence. Moreover, even if several CYP97 sequences described in this work (CYP97H1, CYP97 from *Citrus unshiu* and *Cucurbita maxima* and the published structures CYP97A3 (LUT5) and CYP97C1 (LUT1) from *A. thaliana*) possess some sort of largely α-helical domain N-terminal to the P450 domain, a whole domain from residue R117 to L264 is specific to CYP97H1. Poorly modelled by any software due to lack of existing structural references, this N-terminal extension seems to model in helices clustering around the binding pocket entrance, which could play a role in substrate specificity. As highlighted in the three-dimensional model alignment with the structure of CYP97A3 from *A. thaliana*, known to be a β-ring carotene hydroxylase [10], the N-terminal part of the CYP97 family appears located in front of the membrane. The entrance of the substrate channel allows hydrophobic substrates, buried into the phospholipid layer, to reach the heme catalytic pocket without being in contact with a hydrophilic environment (Figure 2B). This channel is long enough to internalise the entire aliphatic chain of a C40 carotene. During the first hydroxylation, the carotene cyclic extremity reaching the active site is hydroxylated, while the opposite cycle of the molecule is embedded in the distal part of the substrate channel. As the molecule is completely incorporated into the substrate channel, the hypothesis that dihydroxylation occurs via P450 dimerisation, with each P450 reacting with one extremity of the carotene, seems improbable, although enzymatic complexes (such as the heterodimers CYP97A3 and CYP97C1 active on β and ε cycles, respectively) are described in [11]. It is more probable that once the first hydroxylation occurs, the substrate flips and enters again in P450 such that its second not-yet-hydroxylated extremity reaches the active site. In this configuration, the other extremity, which has been hydroxylated, should be accepted into the distal part of the substrate channel, which is the case in CYP97A3*At* or CYP97A4*Os* in the production of zeaxanthin [11,12]. However, could the N-terminal extension of the CYP97H1 act as a hydrophobic cap? Located at the entrance of the substrate channel, this cap could uniquely allow the passage of non-hydroxylated carotene. A monohydroxylated carotene could be prevented from completely entering the substrate channel, due to its distal hydrophilic ring being blocked in the N-terminal hydrophobic cap of the P450.

The third specificity of CYP97H1 concerns the substrate channel. If the overall path is predicted to be conserved between CYP97H1 and CYP97A3 (Figure 2B), three main differences exist between the two channels. The first occurs in the inner residues of the substrate channel. The CYP97A3 substrate channel residues that interact with the substrates have been mapped [10]. They appear strictly conserved in CYP97A4 from *O. sativa.* However, CYP97H1 harbours a set of valines at the entrance of the substrate channel, which present less bulky hydrophobic side chains than the phenylalanine equivalents in CYP97A3. The second specificity of the substrate channel is that CYP97A3 and CYP97C1 from *A. thaliana*, and CYP97B4 from *O. sativa*, harbour a short helix at their entrances; this helix is absent in CYP97H1. Thirdly, the specific extension in the N-terminal of CYP97H1 harbours a substrate channel entrance surrounded by a hydrophobic residue patch, which could act as a gatekeeper for hydroxylated substrates.

### 3.2. The F′-G′ Loop of CYP97H1 Is Shorter Compared to the Ones of CYP97 Members

To analyse the impact of the shorter F′-G′ loop, two types of reversion were built based on CYP97A3*At* (Figure 3A), and then the carotene production in strains expressing these chimera were analysed. CYP97A3*At* is the CYP97 model and has recently been crystallised [10]. It is a carotenoid hydroxylase with major In vivo activity towards the β-ring of γ-carotene (monocyclised lycopene), leading to rubixanthin, and towards the β-ring of α-carotene (β,ε-carotene), leading to zeinoxanthin. CYP97A3*At* also has minor activity on the β-rings of β-carotene (β, β-carotene), leading to β-cryptoxanthin and zeaxanthin [10,11]. The first chimera was produced by inserting the six missing counterpart residues found in CYP97A3*At* into _t_CYP97H1, leading to the chimera _t_CYP97H1_6At_. Some β-cryptoxanthin was detected in the _t_CYP97H1_6At_, showing a tolerance for this small addition but at greatly reduced levels (Figure 3B). The second chimera was formed by replacing the entire 18 residues of the shorter F′-G′ loop of CYP97H1 from residues A411 to D428 with the entire F′-G′ loop of 26 residues from CYP97A3*At*, leading to the chimera _t_CYP97H1_fullAt_. In a strain producing lycopene, γ-carotene and β-carotene, expressing _t_CYP97H1_fullAt_ did not enable the production of rubixanthin, β-cryptoxanthin or zeaxanthin, indicating that the modification via full F′-G′ reversion was too severe. The drastic behaviours of these chimeras indicate that the region is key for the enzyme’s activity and/or its expression.

### 3.3. The N-Terminal Additional Domain Is Required for Full CYP97H1 Activity

To investigate if the N-terminal extension of CYP97H1 acts as a hydrophobic cap, a set of mutants was built with a gradual truncation of the N-terminal cap. The mutants were expressed in the β-carotene strain, and the carotene production, with a focus on β-cryptoxanthin, was quantified. (Figure 4). Compared to the full-length CYP97H1, the deletion of the plastid peptide signal from residue M1 to L116 improved activity in *E. coli*. This peptide signal likely induces an incorrect protein folding in *E. coli*, or an erroneous membrane anchoring. Additional deletions were made targeting the N-terminal extension specific to CYP97H1. Three truncations targeted subdomains L116-M182 and L116-M206, and a more drastic truncation targeted L116-L264, which completely removed the N-terminal cap. These iterative truncations led to a gradual loss of β-cryptoxanthin production, resulting in an inactive enzyme with a more severe deletion. Additionally, an internal deletion of the N-cap of CYP97H1, i.e., a deletion between only L116 and L264, led to a collapse in β-cryptoxanthin production.

As β-carotene is a hydrophobic substrate embedded in the cellular membranes, we tested whether membrane anchoring could compensate for N-terminal truncations. CYP97H1 was fused with the BOV fragment from CYP17a *Bos taurus,* known to improve the expression of P450 in *E. coli* [29]. Firstly, the full-length chimera _BOV_CYP97H1, with an intact N-terminal domain from CYP97H1, was found to be functional (Figure 4B). The chimera deleted from the M1 to L264 residues, i.e., _BOV-delM1-L264_CYP97H1, was then assayed. This deletion annihilated β-cryptoxanthin production, showing that membrane anchoring does not rescue a truncation at the N-terminal domain. In the truncated mutants still producing β-cryptoxanthin, no zeaxanthin was detected, even when using a more sensitive mass spectrometry analysis, meaning that removing the N-terminal cap does not turn CYP97H1 into a dihydroxylase. Nevertheless, it appears that this N-terminal cap is crucial for full CYP97H1 monohydroxylase activity.

Could the N-terminal of CYP97H1 be plugged into biterminal enzymes and shift them into monohydroxylase? In a reverse approach in a heterologous expression in *E. coli*, the activity of CYP973 from *A. thaliana* was assayed on β-carotene. CYP97A3 can be expressed in *E. coli*, purified, and is active on retinal in vitro [10]. Expressing the same protein sequence, i.e., an MBP-His6 tag followed by a tobacco etch virus (TEV) protease cleavage site and CYP97A3 (residues 78 to 595), did not produce detectable amounts of hydroxylated β-carotene. However, we found that the truncated version of the first 77 residues of CYP97A3 leads to production of a small amount of hydroxylated β-carotene in vivo. β-cryptoxanthin and zeaxanthin detection were confirmed by mass spectrometry in the strain ETL154, expressing CYP97A3 (Appendix A). In order to assay if the N-terminal extension of CYP97H1 favoured the activity of CYP97A3 towards the monohydroxylation of β-carotene, the chimera _Nterm97H1_CYP97A3 was constructed by plugging the N-terminal part of CYP97H1 into the globular domain of CYP97A3. Compared to the production obtained with CYP97A3, the chimera _Nterm97H1_CYP97A3 achieved a similar production of β-cryptoxanthin but no detectable production of zeaxanthin. As production was very low, a carotenoid extraction ten times higher was achieved, and the level of β-cryptoxanthin was confirmed to be similar using either CYP97A3 or the chimera _Nterm97H1_CYP97A3. In this larger extraction condition, zeaxanthin was detectable with both enzymatic backgrounds but remained lower using the _Nterm97H1_CYP97A3. This result indicates that the CYP97H1 N-terminal cap can be transposed to another P450 globular domain and tends to favour monohydroxylase activity on β-carotene.

### 3.4. The Substrate Channel Entrance Harbours a Hydrophobic Patch Involved in the Monoterminal Activity of CYP97H1

Complementary to the N-terminal domain analysis, the role of substrate channel polymorphism was also investigated by expressing various N-terminal mutants. The five residues in contact with the substrate in the channel ranging between CYP97H1 and CYP97A3 were swapped into the CYP97H1 sequence, leading to V141F, I159F, V164F, V192I and V302I CYP97H1 mutants. Focusing on the entrance of the channel, the short α-helix present in CYP97A3 was inserted in CYP97H1 leading to the chimeras _K117-A127_CYP97H1 and _M182_-_K117-A127_CYP97H1, the latter of which has a truncated N-terminal starting at methionine 182 to favour a more open substrate channel entrance. Neither β-cryptoxanthin nor zeaxanthin was detected in the three mutants, indicating that this region is key for such activity and unable to accommodate drastic mutations (Figure 5). Focusing on the hydrophobic patch composed of phenylalanine, the single mutants F105A, F108A, F129A and F133A were assayed (Figure 5A). All these single mutants exhibited reduced β-cryptoxanthin production (Figure 5B). A small side production of zeaxanthin was detected with CYP97H1 and the F105A mutant, which was confirmed with mass spectrometry. This impact on β-cryptoxanthin production indicates that either the mutant expression is reduced or that these residues play a role in the entrance of the substrate and are important for enzyme activity. In the F105A mutant, the zeaxanthin:β-cryptoxanthin ratio was displaced in favour of zeaxanthin (10:90 in F105A; 5:95 in CYP97H1), suggesting that opening the substrate channel led to a preference for side dihydroxylase activity.

## 4. Discussion

As metabolic pathways cannot be easily stopped at an intermediate step, the production of intermediate metabolites poses a problem. For example, β-cryptoxanthin, a monohydroxylated β-carotene, is a symmetric molecule where hydroxylation occurs at one terminal while the other remains intact. Only one hydroxylase, the cytochrome P450 CYP97H1, has been described as a monohydroxylase [13]. To decipher the domains involved in this behaviour, protein engineering was conducted to identify the key structural domains for the monohydroxylase activity of CYP97H1. Focusing on the idiosyncratic structure domain of CYP97H1, this enzyme appears to have three specific regions in the CYP97 family: (1) a shorter F′-G′ loop, (2) an N-terminal extension and (3) a hydrophobic patch at the substrate channel entrance. The shortened F′-G′ loop is specific to CYP97H1 among the 102 CYP97 family members (Uniprot). The P450 F′-G′ loop is classically involved in substrate specificity. For example, replacing six residues in the F′-G′ loop of CYP2B11 by their equivalents from the CYP2B6 was sufficient to orientate the chimeric enzyme’s biochemical behaviour towards that of CYP2B6 for the substrates cyclophosphamide and 7-ethoxy-4-trifluoromethylcoumarine [28]. In the case of CYP97H1, the specifically short F′-G′ loop of CYP97H1 did not support a partial or full reversion towards CYP97A3*_At_*, which highlights the sensitivity of this region to the correct P450 expression or activity. A more progressive insertion of the six missing residues could help to identify the breaking point. Gradual truncations in the N-terminal specific to CYP97H1 led to a gradual decrease in β-cryptoxanthin production. When transposed to another globular cytochrome P450 (CYP97A3 from *A. thaliana*), the N-terminal domain from CYP97H1 tended to limit the extent of dihydroxylase activity of the chimera. Several structural studies indicate that P450s tend to cycle between open and closed structures [30]. For example, the internal elasticity of CYP2D6 expands the substrate access channel and thus the substrate specificity [31]. In the case of the additional N-terminal domain of CYP97H1, this extension could modulate the elasticity at the entrance of the substrate channel to play a role in substrate filtering. To understand the inner mechanism embedded inside this N-terminal domain, a mutagenesis study was conducted. Modifying the composition of the phenylalanine patch at the entrance of the predicted substrate channel led to a decrease in β-cryptoxanthin production and, in the case of the F105A mutant, tended to favour dihydroxylase activity. Substrate recognition by surface residues near the access channel has also been observed in other CYPs, such as the T192 and S190 residues of the cytochrome P450cam [32]. Thus, surface residue analysis needs to be taken into account for CYP biochemical properties and should be coupled with in silico substrate channel studies [33]. Taken together, these protein engineering results indicate that the shorter F′-G′ loop structure, the full N-terminal extension and the substrate channel entrance hydrophobic patch play a role in the full activity of CYP97H1, and that the hydrophobic patch from the N-terminal region tends to favour monohydroxylase activity. This N-terminal domain, which is present only in CYP97H1, thus appears to be essential for its full activity and is involved in substrate specificity. In addition to the role of the N-terminal domain in substrate specificity, the implication of CYP97H1 in enzyme complexes should also be discussed. In a *Euglena gracilis* CYP97H1 knockdown mutant, the expressions of geranylgeranyl pyrophosphate synthase (crtE) and phytoene synthase (crtB), both involved in carotenoid precursor biosynthesis, were unchanged, but the total carotenoid content dropped to 10% [13]. This suggests that CYP97H1 plays an essential role in the primary carotenoid biosynthetic pathway. Some cytochrome P450s are known to act in dimers or enzymatic complexes [34]. For example, CYP97A3 and CYP97C1 are thought to work in a protein complex involving two types of lycopene cyclases (ε- and β-cyclases) in *A. thaliana* [11]. In this manner, CYP97H1 could be part of an enzymatic complex including other carotenoid enzymes involved in the upper part of the pathway, and the consequences of its N-terminal original extension could be analysed from this perspective.

## 5. Conclusions and Future Perspectives

Enzymatic substrate specificity remains an open landscape, and can be explored using artificial intelligence and dynamic modelling [35,36]. To assess if external domain additions exist in other CYP families, it would be interesting to compare their primary sequences and isolate the longer ones compared to the classical length of a P450. In a complementary approach, taking advantage of automatic modelling by artificial intelligence could allow large-scale tridimensional alignments and reveal original protuberances coming out from the conserved, prism-like shape of the CYP. Building this in silico library could facilitate the identification of original non-catalytic domains, including in mammal and human CYPs, which would be useful targets for biochemical studies.

Complementary to active site engineering, protein engineering can also focus on the non-catalytic parts of the enzyme, such as its surface, as illustrated with the CYP97H1 N-terminal domain. Indeed, the enzymatic surface is the first part to be in contact with the metabolite; it is involved in substrate entrance and plays a role in the interactions inside enzymatic complexes or with the components of cellular membranes. Thus, taking into account the role of non-catalytic domains can help to decipher original enzymatic behaviours occurring in vivo.

## Figures and Tables

**Figure 1 biomolecules-13-00366-f001:**
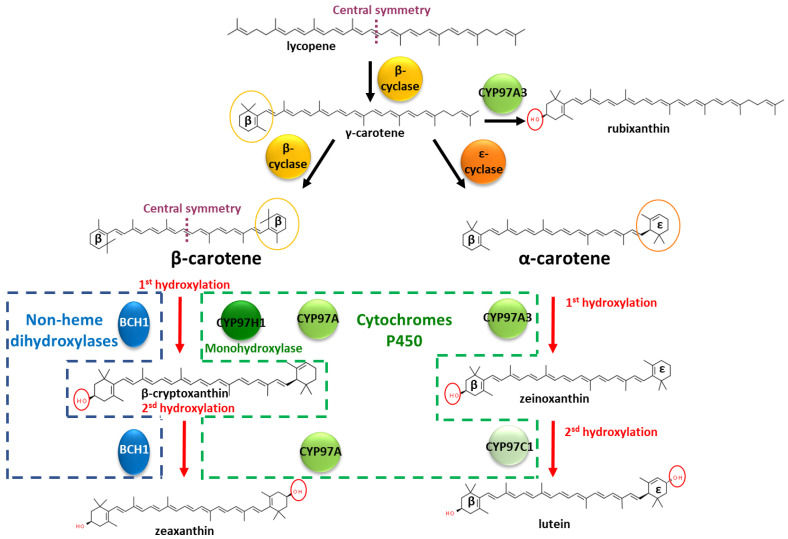
Hydroxylations in the β-carotene and α-carotene pathways with characterised mono- and dihydroxylases. β-cyclase and ε-cyclase refer to lycopene cyclase, leading to β or ε cyclisation, respectively. BCH1 refers to Bacterial Carotene Hydroxylase 1 from *A. thaliana*. CYP97A refers to the cytochrome P450 97 family, clan A. CYP97H1 originates from *E. gracilis*; CYP97A3 and CYP97C1 originate from *A. thaliana*.

**Figure 2 biomolecules-13-00366-f002:**
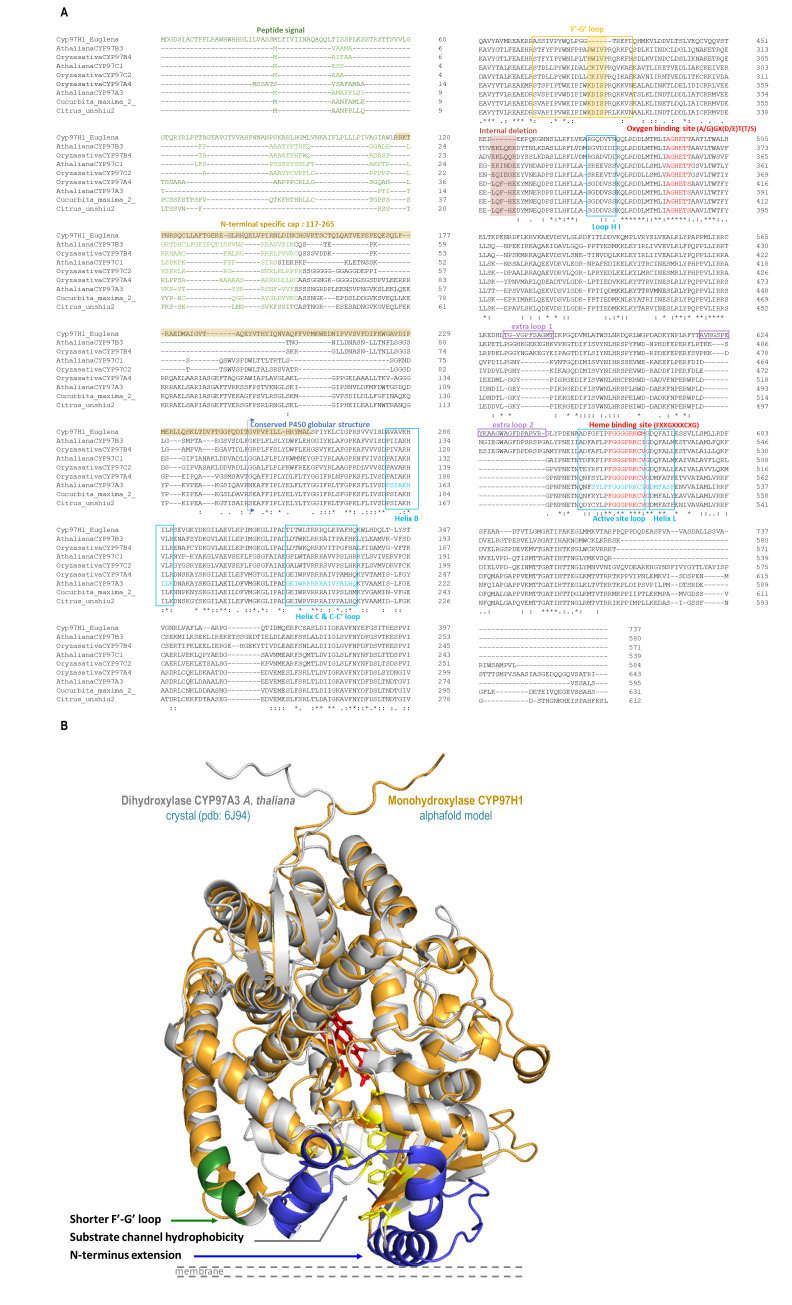
(**A**): CYP97 alignment and key P450s domains of CYP97H1 from *E. gracilis*, CYP97A3, CYP97B3, CYP97C1 from *A. thaliana*, CYP97A4, CYP97B4, CYP97C2 from *O. sativa*, _Ct_CYP97 from *C. unshiu* and CYP97 K15747 from *C. maxima*. (**B**): Tridimensional alignment of CYP97A3 from *A. thaliana* (grey, heme in red) and CYP97H1 from *E. gracilis* (orange). The three specific regions of CYP97H1 are highlighted: F′-G′ loop (green), substrate channel hydrophobicity (yellow) and N-terminal extension (blue).

**Figure 3 biomolecules-13-00366-f003:**
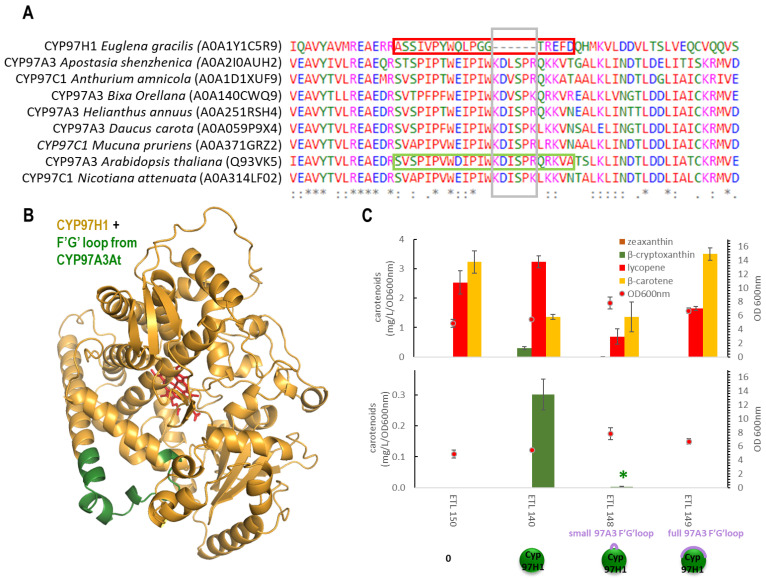
(**A**): CYP97 family alignment with a focus on the F′-G′ region; species and sequence references were extracted from the Uniprot database. (**B**): Alphafold2 modelling of the chimera _t_CYP97H1_fullAt_ (CYP97H1 from *E. gracilis* is in gold, F′-G′ loop from *A. thaliana* is in green, and the heme from CYP97A3 alignment is in red). (**C**): In vivo carotenoid production in the strains expressing F′-G′ loop P450 chimeras.

**Figure 4 biomolecules-13-00366-f004:**
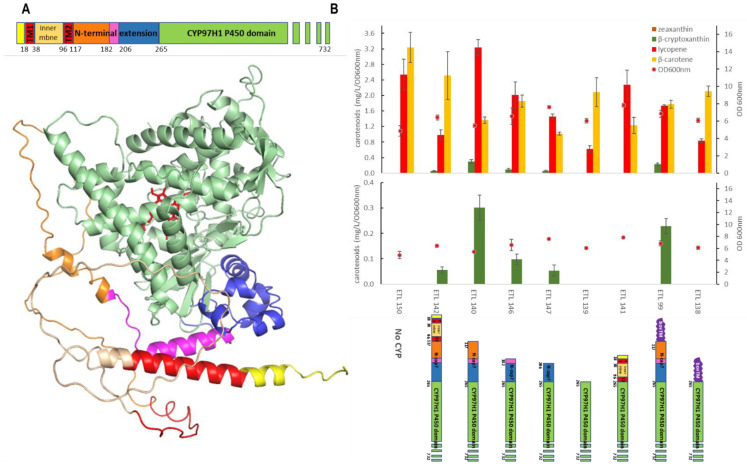
(**A**): Full-length modelling of CYP97H1, with poor modelling of the N-terminal domain. The model was rotated horizontally at 90 °C compared to the CYP97H1 model of Figure 2B. The heme from the CYP97A3 crystal is represented in red. The N-terminal colouring represents the sequential truncated mutants assayed. (**B**): In vivo carotenoid production in strains expressing CYP97H1 N-terminal truncations.

**Figure 5 biomolecules-13-00366-f005:**
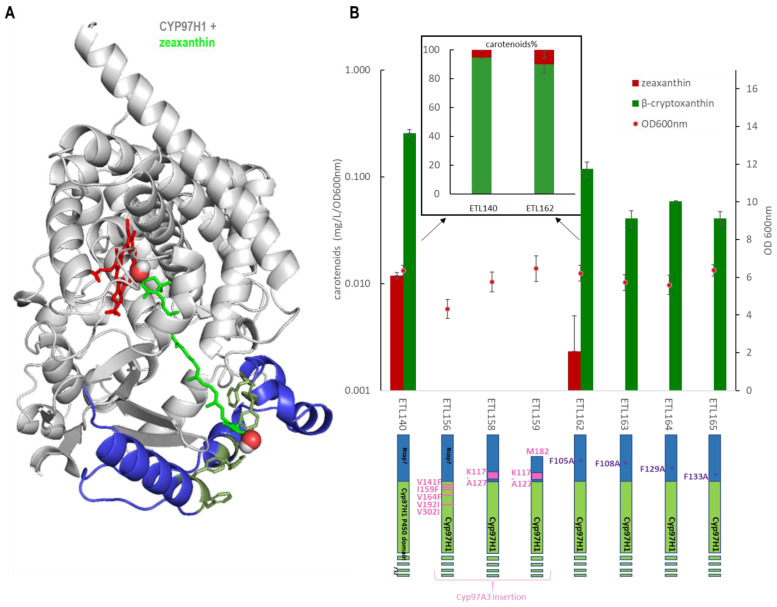
(**A**): Docking of zeaxanthin (green) into CYP97H1. The specific CYP97H1 N-terminal domain (blue) harbours a substrate channel extension surrounded by hydrophobic residues (dark green), where the second hydroxyl group of zeaxanthin colocalizes. The CYP97H1 globular P450 domain (grey) harbours the catalytic heme from CYP97A3 (red) (**B**): In vivo carotenoid production (logarithmic scale) in strains expressing the CYP97H1 chimera with CYP97A3 residues or domains (pink) or with mutations in the hydrophobic patch (purple). **Inset:** ratio of zeaxanthin and β-cryptoxanthin in CYP97H1 and the F105A mutant, averaged over three experiments, with each including four biological replicates.

**Table 1 biomolecules-13-00366-t001:** Strains and plasmids used in this study. The core strain was a ETL90: BL21 adh::T7-GroESL, coexpressing four plasmids: p15A-spec-aroC-hmgS-atoB-hmgR + p15A-cam-aroB-mevK-pmk-pmd-idi + p15A-kan-aroA-IspA-CrtE-CrtB-CrtI+ p15A-amp-CrtY − T7 ferredoxin/ferredoxin reductase *S. oleracea*- T7 CYP. On the fourth “hydroxylase” plasmid, the CYP sequence refers to the different engineered cytochrome P450 sequences as described in Appendix A.

Strain	“Hydroxylase” Plasmid	Specific CYP Sequence	Source
ETL99	pTL45	bov-CYP97H1	this work
ETL138	pTL70	bov-capless-CYP97H1	this work
ETL139	pTL71	supertruncated-CYP97H1	this work
ETL140	pTL72	tCYP97H1	this work
ETL141	pTL73	capless CYP97H1	this work
ETL142	pTL68	full-length CYP97H1	this work
ETL146	pTL79	M182-CYP97H1	this work
ETL147	pTL80	M206-CYP97H1	this work
ETL148	pTL82	tCYP97H1-6At: 6 F′-G′ loop residues from CYP97A3At	this work
ETL149	pTL81	tCYP97H1-fullAt: full F′-G′ loop from CYP97A3At	this work
ETL150	p15A-amp-CrtY	no CYP	Zhang et al., 2018, [14]
ETL154	pTL83	tCYP97A3At	this work
ETL155	pTL84	Nterm97H1-CYP97A3At	this work
ETL156	pTL88	tCYP97H1 with substrate channel from CYP97AAt	this work
ETL158	pTL95	K117-A127At-CYP97H1	this work
ETL159	pTL96	M182-K117-A127At-CYP97H1	this work
ETL160	pTL98	MBP-CYP97A3At	this work
ETL162	pTL91	tCYP97H1 F105A	this work
ETL163	pTL92	tCYP97H1 F108A	this work
ETL164	pTL93	tCYP97H1 F129A	this work
ETL165	pTL94	tCYP97H1 F133A	this work

**Table 2 biomolecules-13-00366-t002:** Confirmation of the mass spectrometry profile of carotenoids using standards.

No.	RT (min)	Compound Name	Molecular Formula	Precursor Ion (*m/z*)	Key Fragments (*m*/*z*)
1	3.45	zeaxanthin	C_40_H_56_O_2_	568.4180	**476.3476**
2	4.99	rubixanthin	C_40_H_56_O	552.4341	**460.3706**, 391.2997
3	5.21	β-cryptoxanthin	C_40_H_56_O	552.4347	**460.3705**
4	8.10	lycopene	C_40_H_56_	536.4164	467.3490, **444.3568**
5	9.45	β-carotene	C_40_H_56_	536.4137	**444.3602**

## Data Availability

All data supporting the findings of this study are available in the article, Appendix A, or upon request from the corresponding author.

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
