# Peer review of "Cytochrome P450 Surface Domains Prevent the β-Carotene Monohydroxylase CYP97H1 of *Euglena gracilis* from Acting as a Dihydroxylase"

_biomolecules, 2023, doi:10.3390/biom13020366_

Round 1
Reviewer 1 Report
In this manuscript, the authors have performed series of experiments and got wonderful results in which they verified and confirmed that CYP domain highlights the role of distal and non-catalytic domains in regulating enzyme specificity. The current study is interesting and has sufficient merit to be considered for publication in this journal. I have no major concerns but following points should be considered before its final consideration:
1) It would be better if the authors describe the experimental conditions of the LC-MS/MS analysis.
2) As the current study is quite interesting. I suggest the authors to separately write the conclusion section and a separate heading of future prospects of this study by summarizing the outcomes of this study on human beings.
3) There are few grammatical mistakes and syntax errors. The whole manuscript needs critical revision to remove all the grammatical mistakes and syntax errors.
Reviewer 2 Report
This elegant study addresses the mechanistic basis of the unique selectivity of CYP97H1 from Euglena gracilis for the monohydroxylation of carotene. The experiments are carefully designed and well performed. The structural and mechanistic analysis is stringent and conclusive. The paper will be of high interest to the researchers in the area of mechanistic enzymology of cytochromes P450. The manuscript is well-written and is nearly ready for publication. I have only a few minor points to mention:
1. Page 2, last sentence of the first paragraph: “particularly from the CYP97 family [5]” - it may be worth adding a sentence or two characterizing the biological role of the family and the occurrence of these enzymes in different species. Some readers may not be familiar with carotene metabolism, and a simple mention of the CYP97 family may not be informative enough for them.
2. Page 2, line 12 from the bottom: “CYP97A3 from A. thaliana or CYP97A4 for Oriza sativa” - shouldn’t it be “from Oriza sativa”?
3. Page 6, line 7 from the bottom: "a set of valine...which are more hydrophilic than the phenylalanine equivalents" – the stating that valine has any degree of hydrophily sounds quite extravagant. It would be more appropriate to say that valine has a less bulky hydrophobic moiety than phenylalanine.
4. I would suggest moving Table S1 to the main text of the article. The identifiers of the E. coli strains that bear different CYP97 constructs, which are defined in this table, are widely used in the manuscript, and the need for reader to refer to the Supplement to decipher these identifiers, and to understand Fig. 3 and Fig. 4 in particular, may be annoying.
Reviewer 3 Report
The authors performed a protein engineering work in CYP97H1 from Euglena gracilis with the aim to determine which enzymatic domains are involved in the regioselectivity, conferring a unique monohydroxylase activity on a substrate offering two identical sites for hydroxylation. In general, the work has scientific soundness and will be of interest to those working with CYP97 dihydroxylases and with hydroxylations in the β-carotene and α-carotene pathways. However, there are some points that must be considered.
1.- The title is ambiguous, it is not reflects the content of the manuscript, as a suggestion, include at least, the name of the protein studied as well as the name of the organism.
2.- In results, please describe first at the begging of each section the results, and then, include the figure, additionally, in some sections, the subsection of the corresponding figure is not cited, only a general reference about the figure is given.
3.- Material and methods section needs to be improved, a limited description about the procedures is given please provide a more detailed information.
4.- In the same context, how the docking of zeaxanthin (Figure 5A) was performed? please include this information in material and methods.
5.- Discussion section seems a resume of the results, a deeper analysis of the data, based on literature, is needed.
6. Please, improve the quality of figure 2A, it is difficult to analyze.
Reviewer 4 Report
This article is one of the best examples of how P450 non-catalytic domains influence substrate specificity or catalytic activity. In the case of CYP71H1 from Euglena gracilis, the authors presented all of the evidence in this regard.
I have only one suggestion for the authors.
The present article title is far too broad. I suggest that authors be precise about the P450, organism, name and which domain plays a role in catalytic activity. I recommend that the authors modify the article title to include the proposed information.
Reviewer 5 Report
In the present work, the authors concentrated on the determination of which cytochrome P450 (CYP97H1) domains are involved in regioselectivity, conferring a unique monohydroxylase activity on a substrate (β-carotene) offering two identical sites for hydroxylation. The research was planned and described with great care. Overall, the manuscript is well thought out. It provides convincing data for the understanding of CYP regioselectivity and is of much interest to the readers. My suggestions are as under:
[Major concern]
· In the Introduction section: The last paragraph contains the obtained results and conclusions. I think that they should be moved to the next part of the work. However, there is no clear indication of the aim of the work and the research methodology used to achieve it.
· There is no Conclusion section. Add also a few sentences about future perspectives.
[Minor concern]
In the Introduction section:
· 8th line (p. 1): Lack of dot after ‘approaches’.
· 2nd paragraph (p. 2): ‘BHC1’ or ‘BCH1’ – two versions appear.
· Enter the CYP designation for cytochrome P450 on first use.
· … CYP97A4 from Oriza sativa.
In Figure 1:
· Change ‘Non heme dihydroxylases’ for ‘Non-heme dihydroxylases’. In the figure caption, explain the meaning of the abbreviations used.
In the Materials and Methods section:
· Please use the consistent form for values and units (e.g., 25 g/L instead of 25g/L, 34 μg/mL instead of 34 μg/ml).
· Include the manufacturer for HPLC and MS/MS systems.
· Zhang et al. 2018, De Ritter et Purcell 1981 – include reference numbers here.
· … consisted of 0.1% formic acid.
In the Results section:
· 1st paragraph (p.6): The last sentence (‘However, could the N-terminal extension of the CYP97H1 act as an hydrophobic cap …) is too long and thus difficult to read. Please rewrite it.
· 2nd paragraph, 2nd line (p. 6): There is a typo in ‘wether’ (
In the Discussion section:
· 3rd paragraph: Look at the phrase ‘… with cellular compounds such as membranes’. Might be better to use ‘components of cellular membranes’?
[Technical remark]
In the future, it would be worth numbering the lines – it would make it easier to check the work.
With kind regards,
Reviewer
Round 2
Reviewer 3 Report
The manuscript was corrected according to the suggestions, therefore, I recommend its publication.
Reviewer 5 Report
Thank you for making the indicated corrections. I believe this manuscript is acceptable in its present form.